# Nourishing the Human Holobiont to Reduce the Risk of Non-Communicable Diseases: A Cow's Milk Evidence Map Example

Rodney R. Dietert [1], Margaret E. Coleman [2,*], D. Warner North [3] and Michele M. Stephenson [4]

[1] Department of Microbiology and Immunology, Cornell University, Ithaca, NY 14853, USA; rrd1@cornell.edu
[2] Coleman Scientific Consulting, Groton, NY 13073, USA
[3] NorthWorks, San Francisco, CA 94133, USA; northworks@mindspring.com
[4] Advancement and External Affairs, Syracuse University, Syracuse, NY 13244, USA; mmstephe@syr.edu
* Correspondence: peg@colemanscientific.org

**Abstract:** The microbiome revolution brought the realization that diet, health, and safety for humans in reality means diet, health, and safety for the human holobiont/superorganism. Eating healthier means much more than just feeding human cells. Our diet must also nourish the combination of our microbiome and our connected physiological systems (e.g., the microimmunosome). For this reason, there has been an interest in returning to ancestral "complete" unprocessed foods enriched in microbes, including raw milks. To contribute to this inevitable "nourishing the holobiont" trend, we introduce a systematic risk–benefit analysis tool (evidence mapping), which facilitates transdisciplinary state-of-the-science decisions that transcend single scientific disciplines. Our prior paper developed an evidence map (a type of risk–benefit mind map) for raw vs. processed/pasteurized human breast milk. In the present paper, we follow with a comprehensive evidence map and narrative for raw/natural vs. processed/pasteurized cow's milk. Importantly, the evidence maps incorporate clinical data for both infectious and non-communicable diseases and allow the impact of modern agricultural, food management, and medical and veterinary monitoring outcomes to be captured. Additionally, we focus on the impact of raw milks (as "complete" foods) on the microimmunosome, the microbiome-systems biology unit that significantly determines risk of the world's number one cause of human death, non-communicable diseases.

**Keywords:** evidence mapping; milk microbiota; risk–benefit analysis; microimmunosome; microbiome; anti-allergy factors; complete foods

## 1. Introduction

Public health centers around the protection of the general healthy population and those more susceptible from adverse effects associated with contaminated food, food additives, drugs, chemicals, and physical agents (e.g., radiation). Among the greatest challenges for public health have been efforts to combat the decades-long, number one global killer, non-communicable diseases (NCDs). However, as illustrated recently by Dietert [1], this has been a losing battle in which institutions such as the World Health Organization have stressed the need to eat healthier [2]. However, what exactly does eating healthier mean in an era where humans are known to be mainly microbial genetically [3], and infants require specific microbiome intervention during the first 1000 days to achieve immune balance across mucosal tissues, reduce impending inflammation [4], and minimize the risk of later-life NCDs (e.g., asthma and obesity) [1,5].

A series of recent reviews, [1,5,6], illustrated the benefits of what have been termed Microbiome-First Approaches for: (1) medicine and safety [1], (2) pain prevention and management [6], and (3) public health protection against both NCDs and pathobionts [1]. In this review, we begin examination of the potential for microbiota-laden foods to aid what has been termed the microimmunosome [7]. This is the systems biology combination of the

microbiome, the barrier (e.g., gut lining), and the underlying immune cells. The majority of all human immune cells are located in the gut microimmunosome [8]. This paper focuses on cow's milk as a major impactor of the microimmunosome and the importance of including the microbiome and immune system in evidence-based, risk–benefit decision making. There are major differences for the microimmunosome when cow's milk is consumed in its unpasteurized (raw) form versus after pasteurization. Additionally, diet is known to be a major driver of microbiome status, and microbiota-laden foods can be a doubly beneficial component supporting human microbiome maintenance and, when needed, re-biosis. This review uses evidence-based mapping of raw versus pasteurized cow's milk to illustrate the importance of putting the microbiome first in health risk decisions.

## 2. Advancing Knowledge of Milk and Milk Microbiota

Milks from healthy mammals are complex living foods that contain all the nutrients needed for offspring to grow and thrive (fats and lipids; simple sugars (predominantly lactose) and oligosaccharides; proteins, peptides, and amino acids; and vitamins and minerals). In addition, many specialized bioactive components of milk function by influencing rates of growth, development and maturation of mammalian tissue systems, and health (growth factors; hormones; enzymes; cytokines and other immunologic factors; and various antibacterial compounds). A key component of the bioactivity of milks is the interdependent networks of microbial communities making up the natural microbiota of milks [9–11].

The microbiota of milks is dense and diverse, and recent reviews describe some core microbes typically present in both human breastmilk and dairy milks [10,12]. The focus of this study is on fresh unprocessed (raw) cow's milk that has not been homogenized, pasteurized, or fermented, as well as pasteurized cow's milk. Certainly, consumers may also include other raw milk products enriched in microbes (e.g., butter, cheese, and kefir) in their diets that may also contribute to benefits and risks. However, the present study focuses on evidence of benefits and risks associated with consumption of fluid raw and pasteurized milks from cows. Additional context on human and cow's milk microbiota is provided in Supplemental Section A.

Bacterial densities exceeding $10^4$ cfu/mL are reported for raw bovine milks from multiple studies [11] and are similar to the magnitudes in human milk. Parente et al. [11] compiled nearly 3000 microbial sequence results on the microbiota of dairy products in a publicly accessible database (FoodMicrobionet; http://www.foodmicrobionet.org/ accessed on 24 April 2020) and reported nearly 2000 taxa identified at the genus level or higher in raw bulk tank milk across five recent bovine microbiota studies [11]. Many of the studies report the top 10 to 50 most prevalent genera amongst hundreds of genera present in the milk microbiota to summarize in tables and figures, including genera these researchers noted as potentially beneficial microbes (*Lactobacillus*, *Streptococcus*, *Lactococcus*, *Staphylococcus*, *Corynebacterium*) [11]. However, many genera are present at densities much less than 1% total abundance [10,11].

Few milk microbiota studies are powered to identify microbes other than predominant taxa to the genus (and species) level by culture-independent methods, and rarely do available studies identify and quantify typical foodborne pathogens (e.g., *Campylobacter* spp., enteropathogenic *E. coli* strains, *Listeria monocytogenes*, and *Salmonella* spp.) in raw milks. Although this limitation also applied to the milk microbiome study of Liu [13] as well, the authors did analyze 16 bovine retail raw milk samples in triplicate for pathogens by culture methods and found none positive. If present, potential pathogens appear to represent extremely small fractions of the microbes present in milks from healthy individuals. In contrast, milk microbiota for individuals with mastitis, inflammation of the mammary glands, is often dominated by 'blooms' of opportunistic pathogens and lower diversity of microbes, suggesting that mastitis may reflect dysbiosis or disruption of the microbiota of healthy mammary tissue [10,11,14–16]. Note that mastitis is the most frequent disease reported in dairy cows [17]. The USDA [18] reported clinical mastitis in nearly 25% of cows sampled in 2013, and less than 5% of mastitic cows died from mammary infections. Similar

rates of 20–25% mastitic infections are reported in studies of lactating women in multiple countries [19].

Extensive literature exists documenting a plethora of interdependent factors from the micro scale to the global scale that influence the microbiota of milks [10,12] (Supplementary file S1). Many of the breastmilk and cow's milk studies cited herein are reviews or systematic reviews [10–12,20] that illustrate the breadth of evidence (some studies consistent, some ambiguous, some conflicting) and the need for further deliberation about evidence for benefits and risks posed by both raw and pasteurized milks.

## 3. Historical Influences of Milk Production Practices on Health and Safety

Mammalian milks are widely recognized as 200-million-year-old superfoods, and domestication of ruminants for human milk consumption likely began in the 9th millennium BC [21]. It would appear from recorded history that humans have consumed raw cow's milk for millennia before pasteurization began in the 20th century.

It is true that consumption of cow's milk was one factor associated with high infant mortality at the turn of the 19th century in the US and around the world. Scholars have attributed the high mortality from that time not to the natural microbes present in raw milk, but in large part to the industrial revolution that enabled profitable but dangerous partnerships between distillers and urban dairies to flourish for many decades in the US and Europe as documented by Egan [22] and Obladen [23]. These studies document that urban populations who could not afford 'country milk' from healthy cows raised on pasture instead consumed adulterated 'swill milk' from unhealthy, diseased, and dying urban cows starved then fed hot distillery waste or 'swill'. 'Swill milk' was reportedly adulterated by addition of water, flour, chalk, plaster of Paris, sugars, salts, bicarbonate of soda, and other compounds to mask its thin, bluish appearance. In addition to high infant mortality associated with human consumption of 'swill milk', these industrialized urban 'dairy' practices were inhumane to cows that reportedly survived only a year in deplorably filthy urban 'stables' that filled feeding troughs with swill insufficient nutritionally to sustain healthy cows [22].

Further, diverse factors at the turn of the century contributed to high mortality rates in cities compared to rural populations, and higher rates of diphtheria and croup, diarrheal illness, consumption or pulmonary tuberculosis, pneumonia, measles and other diseases were associated with overcrowded cities, not rural populations [24]. Contributing factors to disease transmission in cities included the lack of: safe water; reliable systems of sewage and manure disposal; and reliable refrigeration during milk transportation as well as storage in homes [22,23,25]. In addition to being subject to poor-quality, contaminated food and polluted water and environments, the urban poor were also subject to oppressive working conditions and housed in overcrowded unventilated tenements. These factors combined [25,26] to cause a high burden of infectious disease mortality via airborne transmission (including diseases listed above and scarlet fever, whooping cough, and influenza) or via close contact with infected people, as well as via contaminated water (typhoid fever) or food (diarrheal diseases) [22–24,27].

Modern dairy practices in many countries include vaccination of livestock against brucellosis and tuberculosis, proper sanitation and good agricultural practices, Hazard Analysis and Critical Control Point (HACCP) programs, and Test-and-Hold microbial monitoring programs for raw milk [28,29]. Such practices are clearly not representative of the urban 'swill' dairies that persisted within, and later, outside the limits of large cities around the world to meet demand for milk in rapidly growing metropolitan areas in the 19th and 20th centuries.

The evolving context for risk science relevant to raw and pasteurized milks is provided herein from perspectives of immunology, microbiology, and decision science as a starting point to consider evidence of benefits and risks for bovine (cow) milks.

## 4. Regulating the Microimmunosome

The interaction of the immune system with microbes defines biologically what is self and what is not. This interaction is so fundamental for self-definition/integrity that incompatibilities between the immune system and microbiota can separate out new species through a process of self-death by massive inflammation [30,31]. Therefore, establishing a robust microbiome (particularly in the gut) that co-matures with the developing immune system and continues to regulate immune responses throughout life can be the difference between a healthful life versus one filled with pathogen-induced illnesses and/or immune-inflicted inflammation and death.

When considering risk of both NCDs as well as host defenses against pathobionts, the systems biology unit comprising the microbiome, barrier (e.g., gut epithelium layer), and the underlying immune system is of critical importance. The unit has been termed the microimmunosome [7]. The microbiome status is pivotal in determining pathobiont load, the level of antibiotic resistance genes, barrier integrity, and the potential for immune inflammatory pathology (e.g., NCDs). For this reason, seeding and feeding the microbiome to regulate the microimmunosome should be a top priority in both disease prevention as well as comprehensive therapeutic approaches [5].

Foods that naturally provide pre- and/or probiotics (e.g., cow's milk and fermented foods) can have benefit beyond the collection of specific nutrients. Key to understanding and deliberating about the balance of benefits and risks for milks is the need for holistic ecosystem approaches to health that incorporate the advances of 21st-century transdisciplinary knowledge of *Homo sapiens* as 'superorganisms' dependent on our microbiota as our partners in health [32,33]. Cumulative environmental health risks for humans include consumption of foods that neither directly feed keystone microbiota nor replenish the human microbiome.

Thus, a benefit associated with the dense, diverse milk microbiota is gut microbiome 'completeness', 'microbial seeding and feeding' of mammalian gut systems, now understood to contribute to resilience against disturbances that predispose mammals with incomplete microbiota to acute and chronic diseases including obesity, diabetes, and inflammatory diseases [5,33]. Regular exposure to microbes is now understood to be crucial in effectively balancing anti-inflammatory responses to microbes that are tolerated (e.g., commensals and mutualists that dominate the gut microbiota) with pro-inflammatory responses that trigger host defense responses (pathogens in air, food, and water) over a mammal's lifetime. The need for a paradigm shift away from 20th-century ideas about microbes as germs that will kill us, rather than being our partners in health, is consistent with the suggestion regarding the proposed need for considering selection of Recommended Dietary Allowances (RDAs) for microbes in addition to existing RDAs for vitamins [34,35].

Knowledge of the networks of microbes functioning in cooperation, competition, and colonization resistance in dense and diverse ecosystems including raw foods and the host gastrointestinal system is advancing [36–40]. However, quantitative mechanistic data on functionality of specific milk microbes with the gut and immune systems are not available for bovine milk. In fact, these mechanisms likely function synergistically rather than independently in vivo, rendering experimental systems to test single effects overly simplistic representations of the complexities of networked, interdependent interactions observed for the milk microbiota [40].

## 5. Balancing the Immune System via the Microimmunosome

As recently reviewed by Phillips-Farfán et al. [41], the prenatal, neonatal, and infant periods of development are critical for microbiome-immune co-maturation within the microimmunosome. The template can be set for life-long immune responses as directed by signals from the microbiota during early-life critical windows. A plethora of innate and acquired immune cell populations are affected by signals from the microbiome in locations such as the gut. However, for the purpose of distinctions regarding the effects of different

milks, we will focus on microbiota-driven shifts in balance among numerous different mucosal T cell populations.

Five major T cell players in gut include: (i) T helper 1 (Th1) cells which have IFNγ and TNFα as prototype cytokines, drive anti-viral, anti-cancer, and graft rejection responses; (ii) T helper 2 (Th2) cells which have IL-4 and IL-5 as prototype cytokines, drive anti-helminth-parasitic responses but also types of allergic responses; (iii) T helper 17 (Th17) cells which have Il-17A as a prototype cytokine, are critical for mucosal immunity, but when dysregulated drive autoimmunity through misregulated inflammation; (iv) regulatory T (Tregs) cells which can secrete IL-10 and TGFβ, are important for tolerance to self and immune cell homeostasis; and (v) gamma delta (γδ) T cells also called the "unconventional" T cells, because of their distinct receptors. These cells are prominent in the gut and epithelial tissues and seem virtually designed to serve as a communication conduit between the microbiome part of the microimmunosome and human tissue-produced ligands.

Pregnancy and birth are viewed immunologically as predominantly Th2 biased; to reduce risk of miscarriage, Th1 immunity is suppressed temporally. The newborn emerges with a bias toward tolerance but also a default toward inflammation. This Th2 bias and predisposition toward inflammation in newborns must be corrected postnatally by early-life exposure to microbes, including those predominating in the milk microbiota [42,43]. Several investigators have described these postnatal immune changes and the critical impact of breastmilk and its microbiome on early postnatal immune maturation [40,43–46]. The extent to which this microbiota-driven transition happens successfully is a significant factor in both immune status and risk of later childhood and adult disease. Among key microbiota-driven, immune changes that must occur postnatally are: (1) a rebalancing T helper cell function (Th1 vs. Th2 function), (2) polyclonal expansion of gamma-delta T cells, which also develop cytotoxic activity (3) enhanced tolerance via the maturation, expansion, and seeding of the periphery with T regulatory cells (Tregs) achieving a balance between Tregs and Th17 cells, and (4) more effective regulation of macrophages.

Th1 function, driven by interferons and IL-2, supports intracellular immune defense (e.g., responses to invasive bacteria and virus infections), while Th2 function features IL-4, -5, -10, -13, and IgE production and helps control extracellular pathogens (some bacteria and helminth parasites). There is a Th2 bias prenatally, and this must be balanced by an increase in Th1 function shortly after birth. This shift is needed to reduce the risk of specific infectious and chronic immunological diseases. For example, if the prenatal Th2 bias persists during infancy, there is increased risk of atopy and allergic diseases [42]. Breastfeeding facilitates this postnatal immune transition and protects against diseases driven by a Th2 bias [47].

Gamma-Delta T cells are unlike most T cells both in their specific receptor profiles and in their functional responses (which bridge innate and adaptive immunity). While these cells develop early in fetal development, their microbiota-driven, postnatal expansion in the periphery is a feature of neonatal-infant immune maturation. A second change is that these cells postnatally acquire cytotoxic effector functions, that enable them to provide a front-line anti-microbial defense against infections in the baby [48]. Ravens et al. [49] describe how microbiota profiles affect these T cell postnatal processes and result in either a useful balance or disease-promoting dysbiosis.

Tregs play a significant part in postnatal tolerance and the regulation of adaptive immune responses. With human milk, there are at least three components of milk that appear to regulate Tregs: milk microbiota (e.g., Bacteroides) [40], S100 calcium-binding proteins [50], and IgA [51]. In addition to specific milk microbiota, consumption of S100 calcium-binding proteins (ligands for Toll-like receptor 4) from human milk and elevated levels of these proteins in neonates result in: (1) expanded Treg populations in mucosal tissues, (2) enhanced regulation of monocytes and macrophages, (3) improved gut microbiome homeostasis, and (4) reduced risk of diseases/conditions associated with gut microbiota dysbiosis (e.g., childhood obesity) [50]. Overabundance of Tregs in the periphery could result in reduced responses to pathogens. Therefore, balance and control

are important. The challenge is to have sufficient Tregs for tolerance and regulation of inflammation while still having protection from pathogens. Recent findings suggest that this is achieved through the action of IgA in combination with the S100 proteins and milk microbiota. An example for cow's milk and elevated Tregs is discussed in the next section.

## 6. Effects of Raw Cow's Milk on the Microimmunosome and Risk of Allergies and Asthma: Proof of Concept

Asthma and allergic diseases are among the most prevalent childhood-onset NCDs [52,53]. As recently shown by Dietert [5], childhood asthma should be thought of as an entryway NCD that, given present health management, leads to at least 36 additional comorbid NCDs among the ageing asthmatic cohort. These include not only an elevated risk for other allergic diseases but also obesity and metabolic syndrome, neurobehavioral conditions, cancers, cardiovascular diseases, and numerous autoimmune and inflammatory conditions. As a result, there is a premium on reducing the risk for childhood asthma to avoid the elevated risk of later-life multimorbid NCDs with polypharmacy. As we will illustrate, cow's milk offers a perfect and highly relevant example of a naturally, microbial-laden food that can be important during childhood. The status of cow's milk can either significantly protect against asthma and allergic diseases or increase the likelihood of these entryway NCDs. Evidence-based mapping that includes both cow's milk status (processed/pasteurized or unprocessed/raw) and the microbiome-immune interface (the microimmunosome) can capture necessary information to better protect the human holobiont.

In the human infant, introduction of *Bifidobacteria longum* ssp. *infantis* and related communities (a natural component of raw human breast milk) is key to human milk oligosacchrride (HMO) metabolism, difference in bile salt metabolism, changes in gut microbiome profiles, shifts in gut immune cells maturation/balance, reduced enteric inflammation, and protection against asthma and allergic diseases [4,54,55]. These findings support a major microbiome first type of application [1,5,6] for reducing the risk of entryway NCDs (e.g., asthma) that can lead to later-life multimorbidity and polypharmacy. One of the questions is whether raw cow's milk could mediate similar positive changes on the microimmunosome.

A major risk–benefit question concerns whether cow's milk microbiota present in raw milk could have similar positive effects on microbiome composition, immune function and risk of disease. For this reason, the raw milk vs. pasteurized milk comparison across the microimmunosome provides critical information.

Extensive research has been performed on the effects of raw cow's milk microbiota and other components on an ovalbumin-sensitization, food allergy model in mice [56]. In additional studies, Abbring and colleagues [57] demonstrated that skimming of raw cow's milk retained the allergy protective properties. However, heat treatment destroyed that capacity. Loss of the allergy protective activity within cow's milk after heat treatment corresponded to a loss in immunomodulatory activity within the whey protein fraction [58].

Frei and colleagues [59] identified an important immunologic biomarker that is associated with protection against asthma. Children growing up on an animal farm and consuming raw milk produce an antibody against a form of sialic acid (*N*-glycolylneuraminic acid, Neu5Gc). This terminal sugar on glycoproteins and glycolipids is not produced by human cells or the human microbiome. However, it is utilized by both when contributed by other mammals. Raw cow's milk is one of the pathways through which human cells acquire the sugar and then produce antibodies (e.g., IgG) against it. While consumption of raw milk generates these antibodies, pasteurized cow's milk has little capability of stimulating antibody production. The researched suggested that Neu5Gc has anti-inflammatory effects on the human immune system resulting in the suppression of asthma and allergies [59].

Six important finding have emerged from the analysis of cow's milk microbiota and other milk components on the microimmunosome:

1. Immunologically active whey IgG increases the attachment/colonization of *Bifidobacterium longum* ssp. infantis (*B. infantis*) and increases colonization resistance against *Campylobacter jejuni* [60].
2. Raw cow's milk administered in a mouse ovalbumin-sensitized mouse model increases the prevalence of *Lachnospiraceae* UCG-001, *Lachnospiraceae* UCG-008, and *Ruminiclostridium* 5 (Clostridial clusters XIVa and IV) and increases butyrate producers while decreasing inflammation. In contrast, pasteurized cow's milk produces the opposite effect, resulting in microbiome dysbiosis and elevated pro-inflammatory *Proteobacteria.* The protective effect of raw cow's milk against ovalbumin-associated food allergy is heat sensitive. It is associated with immunological changes including a reduction in allergen specific Th2 cells responsiveness and an increase in Treg activity [56].
3. Cow's milk oligosaccharrides in combination with *B. infantis*, can correct diet-associated microbial dysbiosis, reduce gut permeability, and reduce inflammation [61].
4. Raw cow's milk prevents airway inflammation from developing in a mouse model of house dust mite-induced asthma [62].
5. Using a mouse animal model, raw cow's milk but not processed milk appears to have epigenetic effects on FoxP3+ T regulatory cells resulting in reduced allergic symptoms [63].
6. A biomarker, Neu5Gc and antibodies produced against it, are associated with consumption of raw cow's milk. This is a useful biomarker for predicting protection against allergies and asthma [59].

## 7. Approach for Creating the Evidence Map for Cow's Milk

The process for creating a cow's milk evidence-based map for decision making paralleled our recent publication (in this Special Issue): an example comparing pasteurized donor milk and unpasteurized (raw) human breastmilk [64]. As such, the extensive literature on the microbiota of cow's milk and assessments of benefits and risks was compiled over multiple years of this project, including results of searches, as well as manuscripts and reports submitted by our project partners. Combinations of key words (e.g., raw or unpasteurized, milk, microbiota, benefit, risk, bovine OR cow) were used in searches of PubMed, Google Scholar, and the Cochrane database of systematic reviews in combination with discipline-specific key words by multiple authors. Additional studies were identified by forward searching studies citing key references by discipline. This study focuses on recent reviews and human experimental and observational studies.

Wiedemann and colleagues [65] described the structural template though which two potentially opposing views are compared side by side. This evidence comparison increases reliability and transparency for resolving mixed messages in risk–benefit analysis. For example, claims that raw cow's milk is 'inherently dangerous' can invoke fear and dread. However, these claims may or may not be based on evidence from relevant peer-reviewed studies. To better address the potential for lingering adverse risk perceptions based on outdated dogmas concerning commensal microbes and pathobionts, we prepared a qualitative evidence map for the cow's milk ecosystem (Figure 1). The side-by-side comparisons within the evidence map can aid the decision-making using the latest science rather than decades old or generational perceptions embedded in bygone times [66].

The cow's milk evidence map in Figure 1 includes a blue text box for a 'Pro-Argument' about benefits and a 'Contra-Argument' about risks. Human experimental evidence from randomized controlled clinical trials and observational studies, as well as systematic reviews (SRs), meta-analyses (MAs), quantitative microbial risk assessments (QMRAs), and outbreak investigations (OIs), are introduced under the center section associated with the 'Pro-' and 'Contra-Arguments'. Supplemental studies on plausible mechanisms for benefits and risks that attenuate and/or support the 'Pro-' and 'Contra-Arguments' are listed in green beveled-edge boxes on the left section of the evidence maps, with summary information for each study provided in Supplementary Tables S1 and S2. On the right

section of the evidence maps are brief summaries about the evidence basis, conclusions for benefit and risk arguments, and remaining uncertainties.

This cow's milk evidence map for the milk microbiota and other milk components identifies not only the current knowledge but also the gaps that exist for optimized risk–benefit analysis. As such, this serves as a starting point for future steps of the analytic-deliberative process planned for this project.

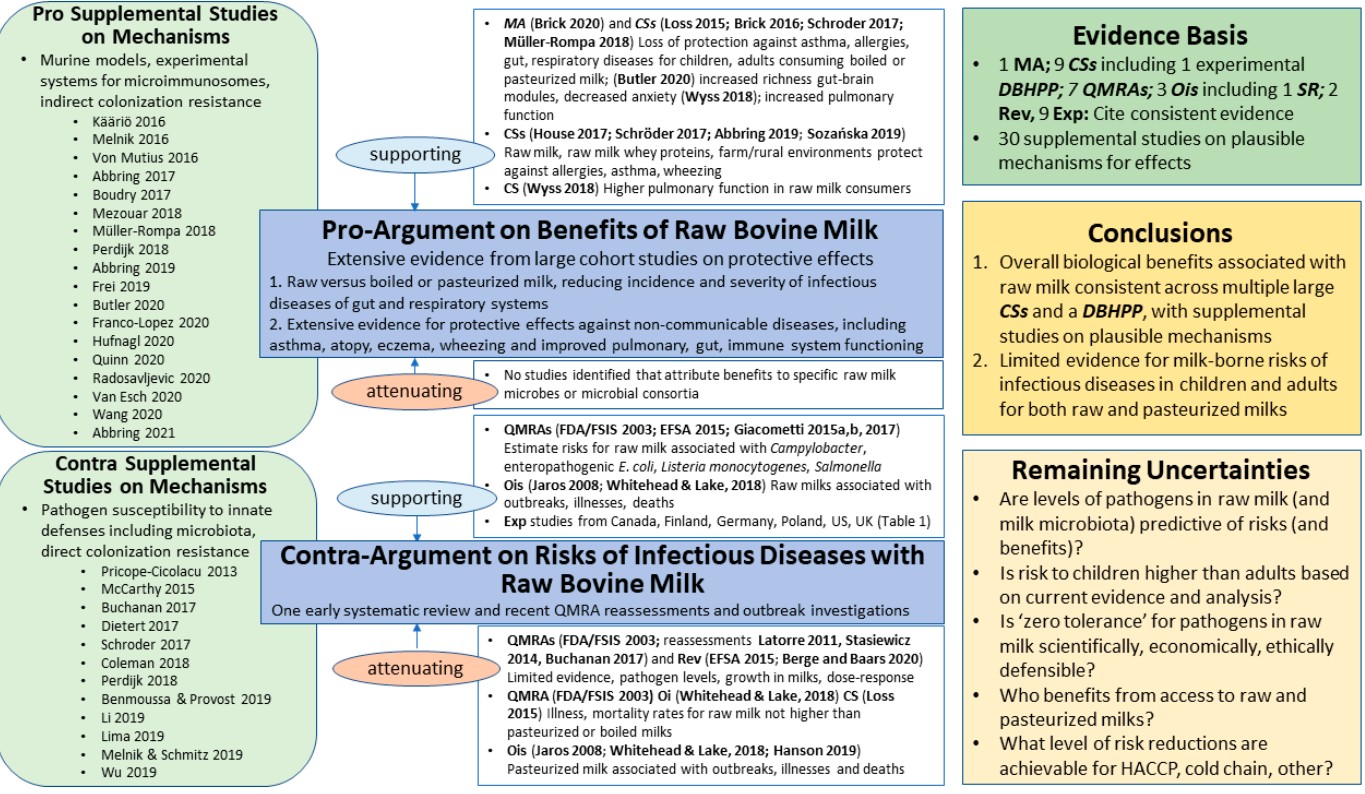

**Figure 1.** Evidence Map for Bovine Milk Ecosystem. CS = human cohort study; DBHPP = double-blinded randomized human provocation pilot; MA = meta analysis; Oi = Outbreak investigation; QMRA = Quantitative Microbial Risk Assessment; Rev = recent review; Exp = exposure study on prevalence of major foodborne pathogens from routine monitoring of licensed raw milk farms; SR = systematic review. See Results section for further explanation of Remaining Uncertainties and Supplemental Materials for further documentation on mechanisms and full references for mechanistic studies. Studies included in the Evidence Basis are listed in the figure by first or first and second author(s) and year, followed by the reference numbers cited in bibliography. The caption lists these studies chronologically, ending with most recent studies: FDA/FSIS, 2003, [67]; Jaros 2008 [68]; Latorre 2011 [69]; Stasiewicz 2014 [70]; EFSA 2015 [71]; Giacometti 2015a,b [72,73]; Loss 2015 [74]; Brick 2016 [75]; Giacometti 2017 [76]; House 2017 [77]; Schröder 2017 [78]; Müller-Rompa 2018 [79]; Whitehead and Lake 2018 [29]; Wyss 2018 [80]; Abbring 2019 [81]; Hanson 2019 [82]; Sozańska 2019 [83]; Berge and Baars 2020 [84]; Brick 2020 [85]; Butler 2020 [86]. For Supplemental Studies on Mechanisms, first or first and second author(s) and year are listed within the figure, and full references provided in Supplementary Tables S1 and S2. See Table 1 for information and reference numbers from nine recent exposure studies.

**Table 1.** Detection of Potential Pathogens in Raw Milk from Recent Studies and Monitoring Programs.

| Country (Reference) | Dates (State if US) | *Campylobacter* | *E. coli* O157:H7 or EHECs | *L. monocytogenes* | *Salmonella* |
|---|---|---|---|---|---|
| Canada (BCHA, 2021; website listed above) | 2015–2021 | 0/192 | 0/192 | 0/192 | 0/192 |
| Poland (Andrzejewska et al., 2019 [87]) | 2014–2018 | 0/113 vending machines; 26/221 (12%) *C. jejuni*, directly from farmers | Not Tested | Not Tested | Not Tested |
| UK (McLauchlin et al., 2020 [88]) | 2017–2019 | 18/635 (2.8%) | 0/58 O157; 3/304 EHEC (0%, 1%) | 1/642 (0.2%) | 3/622 (0.5%) |
| UK (Willis et al., 2018 [89]) | 2014–2016 (routine monitoring) | 2/770 (<0.01%) | 2/770 (<0.01%) | 2/770 >100 cfu/mL (<0.01%) | 0/770 |
| US State Monitoring (database of FOIA source data from licensed farms; Stephenson and Coleman, 2021 [90]) | 2009–2014 (CA) | 0/61 | 0/61 | 0/61 | 0/61 |
| | 2009–2014 (NY) | 6/783 (0.7%) | 0/782 | 1/781 (0.1%) | 0/780 |
| | 2009–2014 (TX) | 4/601 (0.7%) | 0/596 | 4/596 (0.7%) | 11/606 (1.8%) |
| | 2012–2015 (WA) | 0/497 | 0/502 2/501 (0.4%) | 0/502 | 0/494 |
| Germany (Berge & Baars, 2020 [84]) | 2001–2015 (VZM) | 7/2352 (0.3%) | 17/2737 (0.7%) | 30/2999 (1%) | 0/3367 |
| Germany (Berge & Baars, 2020 [84]) | 2001–2015 (not for direct consumption raw, pre-pasteurized) | 17/2258 (0.8%) | 82/5433 (1.5%) | 52/2355 (2.2%) | 0/1084 |
| Finland (Castro et al., 2017 [91]) | 2013–2015 | Not Tested | Not Tested | 5/105 retail bottles (4.8%) 2/115 bulk tanks (1.7%) | Not Tested |
| Finland (Jaakkonen et al., 2019 [92]) | 2014–2015 | 0/789 | 0/789 O157:H7; 2/789 O121:H19 (<1%) | Not Tested | Not Tested |
| US (Del Collo et al., 2017, [93]) | 2014 (17 states) | 13/234 culture; 27/234 PCR (6%; 12%) | Not Tested | Not Tested | Not Tested |
| OVERALL PERCENTAGE POSITIVE | | 93/9740 (0.01%) | 26/10,934 (<0.01%) | 40/9118 (<0.01%) | 14/7976 (<0.01%) |

## 8. Results for Cow's Milk Evidence Mapping

An evidence map is presented for the raw bovine milk ecosystem (Figure 1), including a Pro-Argument on benefits of raw milk and a Contra-Argument on risks of raw milk. For each argument, highlights of 30 studies (systematic reviews, meta-analyses, cohort studies, outbreak investigations, reviews, and exposure studies) are provided that support or attenuate each argument.

Supplemental studies that provide evidence of potential mechanisms for benefits and risks are introduced in Figure 1 (and Supplementary Tables S1 and S2). The categories for potential mechanisms for the breastmilk [64] and bovine milk ecosystems (Figure 1) are similar, though few supporting studies test milks representing both ecosystems.

Due to the high diversity of the milk microbiota in mammals and variability for conditions influencing its composition and abundance (Supplementary file S1) ambiguity of results and conflicting studies are to be expected. Thus, the body of evidence presented in this section illustrates the 'state of the science' and uncertainties for consideration in future cycles of analysis and deliberation (analytic-deliberative process) proposed in the Future Direction section. Supplementary Tables S1 and S2 summarizes relevance and provides full references for the 30 supplemental studies on mechanisms for benefits and risks listed in Figure 1.

### 8.1. Benefits: Pro-Argument

8.1.1. Supporting

Brick and colleagues [85] conducted a meta-analysis that included eight studies in farm children consuming raw or processed milks. The meta-analysis corroborated the protective effect of raw milk consumption early in life (<1 to 5 years) on asthma, current wheeze, hay fever or allergic rhinitis, and atopic sensitization. The effect particularly on asthma was observed not only in children raised on farms but also in children living in rural areas but not on a farm. This demonstrates that the effect of farm milk consumption is independent of other farm exposures and that children not living on a farm can theoretically profit from the protective effects of raw milk consumption.

Butler and colleagues [86] conducted a small observational study (22 participants) that measured aspects of diet, nutrition, and gut and mental health prior to and following a 12 week exposure to organic raw dairy products from grass-fed (pastured) cows. The residential study measured dietary intakes of various foods including raw dairy and documented statistically significant increases in lactobacilli in the gut associated with increasing consumption of raw dairy. In addition, after 12 week exposure to raw dairy, statistically significant increases were measured in fecal short-chain fatty acids (SCFAs proprionate and valerate), intakes of vitamins A and B-12, increase in the functional richness of the gut microbiome profile, as determined by measuring the predictive neuroactive potential using a gut–brain module approach, and significant decrease in anxiety and stress markers for those with higher than median scores on hospital anxiety and depression scale, anxiety subscale (HADS-A).

Abbring and colleagues [81] demonstrated that organic raw milk and native whey proteins in raw milk have lower allergenicity in mouse models of allergy, and lower allergenicity of raw milks was confirmed in children with cow's milk allergies. Eleven children were included in a double-blind randomized proof-of-concept provocation pilot study conducted at University of Kassel, Germany. Children allergic to cow's milk tolerated up to 50 mL of organic raw milk, whereas only 8.6 mL commercially processed (shop) milks (pasteurized and homogenized) were tolerated ($p = 0.0078$). The study demonstrates the immunologic mechanisms (sensitizing capacity; capacity to bind IgE antibodies and induce allergic reaction) accounting for the negative influences of milk processing on the allergenicity of milk. Both allergen specific IgE levels and Th2 cytokine concentrations were inhibited in the murine model. The allergenic effects were suggested to follow time and temperature dependencies, e.g., more severe processing conditions induced greater allergenic effects. The authors note that the whey protein fraction of raw milk contains

multiple immunomodulatory components (immunoglobulins, TGF-β, IL-10, lactoferrin, lysozyme, osteopontin, and lactoperoxidase) known to enhance mucosal barrier function and modulate mucosal immune responses. Mechanisms of protection were consistent for human and murine subjects.

University researcher and pediatrician Professor Barbara Sozańska (Wroclaw Medical University, Wroclaw, Poland) reviewed the extensive literature for large cross-sectional surveys and cohort studies from the European Union (EU) from 2006 to 2018 [83]. Sozańska reviewed both studies on protective effects of farming/non-farming environments and raw/commercial (homogenized and pasteurized)/boiled milks on allergy and asthma in children and adults, as well as potential mechanisms of protection. The following major differences are noted between raw and heated milks: proteins, particularly enzymes; fat and fatty acids; milk microbiota; human and bovine milk exozomes enriched in raw milk, decreased in pasteurized, linked to immune regulatory functions; genetics, epigenetics, raw cow's milk exposure. The study notes a 'debate' from 'skeptics' and 'enthusiasts' about attribution of protection based on ambiguity of the evidence. However, the researcher concludes with certainty that components of raw milk can influence immune function and that raw milks present a promising path for study of allergy prevention.

Müller-Rompa and colleagues [79] reported that asthma and atopy were inversely associated with the presence of a farm within a radius of 100 m of residences for 2265 children in the GABRIELLA cohort with geocoded addresses available. Results include protection from asthma for 1349 non-farm children via consumption of farm milk (raw) and broader diversity of microbial exposures on or near traditional farms for both farm and non-farm children. Protection was completely explained by consumption of farm milk for non-farm children, and was a strong protective factor for farm children, particularly those with another traditional farm nearby.

Wyss and colleagues [80] reported significantly higher lung function for adults in the US (3061 adults in the Agricultural Lung Health Study or ALHS) who had early-life farm exposures including consumption of raw milk as children. The measures of lung function in this study were Forced Expiratory Volume or FEV ($p = 0.04$), Forced Vital Capacity or FVC ($p = 0.01$), and the ratio of these metrics. Raw milk associations were more common in 1936 non-asthmatics than 1125 asthmatics ($p = 0.07$) and linked to beneficial effects on lung growth. Two large European studies were cited reporting higher lung function associated with being born or raised on a farm, consistent with this US study. No other farming variables tested were correlated to higher lung function.

Schröder and colleagues [78] reported significant decreases in regulatory T cells (Tregs) with high farm milk and animal-stable exposure and increased with asthmatics for 111 children enrolled in the longitudinal PASTURE/EFRAIM birth cohort ($p = 0.045$, 56 farm and 36 reference children). The study identified a critical developmental window between 4.5 and 6 years of age for Treg-mediated asthma protection via exposure to raw milk and farm environments.

House and colleagues [77] reported exposure to the farming environment in utero and in early childhood (including raw milk consumption) was strongly associated with reduced risk of allergic sensitization measured by allergen-specific immunoglobulin Es or IgEs (atopic, n = 578, non-atopic n = 2526) for adults in the US (3301 adults in the ALHS). Little or no associations were detected for asthma.

The two studies summarized below [74,75] based on the Protection Against Allergy—Study in Rural Environments (PASTURE) prospective (longitudinal) birth cohort (1133 children living in rural areas of 5 European countries) were included in the Sozańska review [83]. These studies compared different bovine milks (farm milks (raw and boiled); and commercial 'shop' milks (centrifuged, homogenized and pasteurized or ultraheat treated (UHT)). Brick and colleagues [75] conducted a nested case control study based on 84 children at 6 years of age (35 asthmatic and 49 non-asthmatic). Fatty acids were quantified in farm and commercial milk samples for 42 children. Risk of childhood-onset asthma were reduced by previous consumption and continuous consumption of unprocessed (raw) farm milk

compared with commercial milks ($p = 0.01$). Part of the effect was explained by higher fat content of farm milk, particularly higher levels of ω-3 polyunsaturated fatty acids linked to immune balance and lower inflammation, and lack of thermal processing that affects thermolabile microbiota, whey proteins, microRNA and exosomes. Recent consumption of farm milk was more relevant to the protective effect against asthma than consumption of farm milk in the first year of life, providing evidence for continuing effects past infancy.

Loss and colleagues [74] reported consumption for different bovine milks and metrics for immune function (high-sensitivity C-reactive protein (hsCRP) in infant blood samples) and disease (occurrence rhinitis, respiratory tract infections (RTI), and otitis (ear inflammation or infection) in 983 children age 12 months. Raw bovine milk was associated with significantly lower respiratory infections (range of $p$ values from Table 2 for crude associations, 0.011, 0.028, <0.001, and 0.026) in comparison to UHT milk, while pasteurized milk provided no significant protection at $p = 0.05$ level. Protective effects for raw milk remained significant with adjustment for effect of breastfeeding. The protective effects of breastmilk and raw bovine milk were comparable, suggesting similar anti-infective properties of raw milks from human and bovine origin. No clear association between bovine milks and diarrhea was observed, in contrast to concerns about presence of typical foodborne pathogens in raw milks from human and bovine origin. An inverse association was observed for children drinking raw milk and hsCRP value, implying a sustained anti-inflammatory effect for raw bovine milk that has been related to obesity, respiratory impairment, asthma severity and atherogenic lipid profiles. Some protection was also provided against fever by raw, boiled, and pasteurized milks as assessed by weekly health diaries. Early-life consumption of raw farm milk significantly reduced risk of respiratory infections by approximately 30%. Respiratory infections are a major cause of morbidity and mortality worldwide, especially among children, and respiratory infections early in life have been linked to development of chronic diseases such as asthma and non-communicable inflammatory diseases in this and subsequent studies.

**Table 2.** Results Reported by Wu and Colleagues [94] in Study Abstract.

---

- In rumen fluid, feces, bedding, and water at both farms, the most abundant taxa were: *Lactobacillaceae, Prevotellaceae, Ruminococcaceae, Ruminococcaceae*, and *Lactobacillaceae*.

---

- At farm 1, the most abundant taxon in milk and airborne dust was *Aerococcaceae*.
- At farm 2, *Staphylococcaceae* and *Lactobacillaceae* were the most abundant taxa in milk and airborne dust.

---

- At farm 1, the three most prevalent taxa shared between milk and airborne dust microbiota were *Aerococcaceae, Staphylococcaceae*, and *Ruminococcaceae*.
- At farm 2, *Staphylococcaceae, Lactobacillaceae*, and *Ruminococcaceae* were shared between milk and airborne dust microbiota.

---

- The results from SourceTracker indicated that milk microbiota was related to the microbiota from airborne dust.

---

- Using hierarchical clustering and canonical analysis of principal coordinates, it was demonstrated that milk microbiota was associated with bedding microbiota but clearly separated from feed, rumen fluid, feces, and water microbiota.

---

### 8.1.2. Attenuating

No studies were identified that attributed health benefits to specific raw milk microbes or microbial consortia, independent of other bioactive factors present in raw milks.

### 8.2. Risks: Contra-Argument

### 8.2.1. Supporting

Multiple sources of data from microbiology to epidemiology and statistics can provide inputs to risk assessment and inform simulations for comparing alternative strategies for reducing risks.

Data from epidemiologic investigations of outbreaks are generally correlative (identifying potential associations). No published systematic reviews of outbreak reports on raw and pasteurized milks were identified in our repeated searches of the peer reviewed literature.

The most recent study of epidemiologic data in the US [29] analyzed a dataset from the US CDC National Outbreak Reporting System (NORS) database for dairy outbreaks reported from 2005 to 2016 and summarized outbreaks of illness, hospitalizations, and mortality for raw and pasteurized milks and dairy products. Whitehead and Lake (authors' Table 1) [29] reported the following numbers of outbreaks, illnesses, hospitalizations, deaths, and deaths/1000 illnesses, respectively, for raw milk (152, 1735, 176, 2, and 1.2) and for pasteurized milk (6, 1903, 20, 4, and 2.1) in this time period. Whitehead and Lake [29] reported that raw milk was associated with 2 deaths in the 12 year time period assessed, both in cases with significant co-morbidities, one with campylobacteriosis, the other with listeriosis. Different time periods for the same NORS data source considered in earlier studies also included illnesses, hospitalizations, and deaths associated with raw and pasteurized milks, though statistical testing of specific hypotheses about raw and pasteurized milks was limited.

A consensus framework for QMRAs for infectious pathogens [95,96] is organized around four elements as illustrated in our companion manuscript [97]: Hazard Identification; Exposure Assessment; Dose–Response Assessment or Hazard Characterization; and Risk Characterization. This framework is not applicable to assessment of NCDs. While the framework is amenable to inclusion of transdisciplinary quantitative data for each element, assumptions and simple default models are often applied in the absence of rigorous data, particularly for Exposure Assessment and Dose–Response Assessment. Reliance on assumptions lacking valid independent scientific data can thus bias QMRAs and lead to overestimations of risk and underestimations of uncertainties.

Giacometti and colleagues [76] conducted a QMRA for the enteropathogen *E. coli* O157:H7 (also abbreviated EHEC or STEC) that can infect and progress to cause hemolytic uremic syndrome (HUS) associated with severe and sometimes fatal effects, particularly in young children. Over a 7 year period in Italy (2008–2015), 7 of 64 cases of HUS cases were associated with raw milk. Exposure data included very low prevalence and levels (<0.4 MPN/mL) for raw milks from vending machines in Italy. A non-threshold, linear low dose model for Dose–Response Assessment based on epidemiologic investigations of a ground beef outbreak and simulated cooking for contaminated ground beef was applied to estimate risks of HUS for raw milk from vending machines. The authors estimated average annual risk of $8.25 \times 10^{-7}$ for children under 15 years of age, assuming 99% of consumers boil raw milk before consumption, in line with observed HUS cases in this time period associated with raw milk (7 of 64 total HUS cases). Otherwise, the baseline models for lower consumer boiling patterns were much higher than observed cases in the same time period.

Giacometti and colleagues [72] detected *Campylobacter* at low prevalence in raw milk from vending machines in Italy (53 of 15,282 prevalence), and data were too sparse or lacking for estimating a distribution of pathogen levels and an unbiased dose–response model. Estimates of campylobacteriosis cases per year varied widely (zero cases to 8110 cases per 100,000), largely dependent on assumptions for dose–response modeling and consumer behavior.

Giacometti and colleagues [73] estimated risks of listeriosis and salmonellosis for raw milk from vending machines in Italy. No listeriosis cases were predicted per year for both best- and worst-case assumptions; the researchers concluded that the probability of listeriosis linked to consumption of raw milk is low, and listeriosis cases are highly improbable for simulated consumption of raw milk. In contrast, salmonellosis risk estimates varied widely (no cases to 421,454 cases), largely dependent on assumptions for dose–response modeling and consumer behavior (as noted for campylobacteriosis in their 2017 QMRA for HUS). In addition, storage times and temperatures of raw milk were a third source of variability and uncertainty for salmonellosis.

An unpublished report by academic researchers [68] documents criteria for evaluating causality in a detailed systematic review of dairy outbreaks from around the world. Many epidemiologic studies were excluded by Jaros and colleagues because of confounding

factors or unacceptable internal validity. Unfortunately, the authors pooled studies for raw milks with information about other dairy products, rather than reporting fluid raw milk outbreaks separately. Even so, no strong causal association was documented for raw dairy products for any of the pathogens considered. These authors also reported: moderate associations for *Campylobacter*, *E. coli*, *Listeria monocytogenes*, and *Salmonella*; weak association for *Brucella*; and, insufficient evidence for any causal association for pathogens including *Shigella*, *Staphylococcus*, *Streptococcus*, and *Yersinia*.

An early QMRA [67] estimated that listeriosis risks for raw milks were high. For example, the estimated annual risks associated with raw milks were 3.1 serious annual illnesses/deaths, corresponding to risks per serving of $7 \times 10^{-99}$ respectively.

Recent studies depicting exposure (presence/absence) of the major four pathogens in raw milk from licensed farms are summarized in Table 1. The table summarizes data from published studies and a Microsoft Access® database that includes data from US State monitoring (CA, NY, and WA, provided under the US Freedom of Information Act [90] and independent laboratories (provided by British Columbia Herdshare (as of February 2021). The certified laboratory MB Laboratories (Sidney, BC, Canada) conducted analyses of raw milk for the 'BC Fresh Milk Project' of the British Columbia Herdshare Association (BCHA). Readers can review individual laboratory reports for each of 192 samples analyzed to date at https://drive.google.com/drive/folders/0Bz2kJcZ3EjElekV1RmRhMmhBQzg accessed on 15 January 2021. Published studies include multiple years of routine monitoring of licensed farms in Finland, Germany, Poland, the UK, and the US. Overall, the major foodborne pathogens are detected in raw milks produced for direct human consumption without heat treatment at low prevalence (≤0.01%). In other words, more than 99% of 8000 to 11,000 samples of raw milk from routine testing were negative for the major foodborne pathogens.

### 8.2.2. Attenuating

Although enteropathogens may be present in raw milk, the prevalence in raw milk produced for direct human consumption from multiple countries is low, generally below the detection limit or 0.01% positive or less (Table 1). However, in addition, robust QMRAs require additional data in order to estimate exposure to consumers: the levels or counts of potential pathogens per volume or per serving; and rates of decline in competition with the natural milk microbiota at refrigeration temperatures (4.4 °C or 40 °F).

Giacometti and colleagues [76] concluded from their most recent QMRA that the prevalence and levels of STECs in raw milk samples from vending machines were low (38 of 23,752 samples, <1.4 most probable number or MPN/mL), and that the overall public health relevance of STECs and HUS in raw milk is likely to have subsided to negligible levels in Italy (2, 3, 1, 0, 1, 0, and 0 reported annual cases associated with raw milk from 2008 through 2014). The authors note, 'Plainly, this model oversimplifies the complex overall raw milk consumption scenario, based on many assumptions, hypotheses and consumer behavior.' Overall, the available data were consistent with low risk. Other sources (animal contacts, fecal and environmental contacts, and other foods) appear to cause more HUS than raw milk.

Giacometti and colleagues [76] also stated that their previous studies likely overestimated risks for campylobacteriosis [72], listeriosis and salmonellosis [73]. In addition, the latter study [73] noted that the number of salmonellosis cases predicted in their QMRA simulation models exceeded observed numbers of salmonellosis cases in Italy during the same time period by approximately 2 orders of magnitude (3334 to 6662 total salmonellosis cases reported versus 421,454 simulated cases). Thus, significant uncertainties and assumptions for all three QMRAs appear to drive gross overestimates of risk relative to surveillance data available for the same time periods.

A review of QMRAs for raw milk by the European Food Safety Authority (EFSA, pg. 4) [71] provided the following perspective for listeriosis in monitoring programs for raw milk.

Although *L. monocytogenes* is not considered to be one of the main hazards associated with RDM [raw drinking milk] in the EU, the reviewed QMRAs from outside the EU do show that the risk associated with *L. monocytogenes* in raw cow's milk can be mitigated and reduced significantly if the cold chain is well controlled, the shelf-life of raw milk is limited to a few days and there is consumer compliance with these measures/controls.

The statement above from EFSA is also true for the remaining major pathogens (*Campylobacter* spp., EHECs, and *Salmonella* spp.) that cannot outcompete the natural microbiota at refrigeration temperatures [98]. Given appropriate hygienic programs, no recent scientific evidence exists, to our knowledge, that demonstrates conclusively that raw milk is inherently dangerous even though the presence of potential pathogens in raw milk is possible.

Further, assessors in Europe [71] did not find evidence that *Listeria monocytogenes* was important for raw milk risk in the European Union based on evidence from epidemiologic studies and occurrence in EU bulk milk tanks and raw (drinking) milk, as well as expert opinion. This finding is consistent with the predictions for listeriosis risk [73]. EFSA also concluded that outbreak data are insufficient for risk evaluation.

A more recent review by Berge and Baars [84] cited studies on the influence of hygienic practices including HACCP, as well as results of microbiologic testing of raw milks from Germany, with supplemental testing by Robert Koch Institute consistent for Vorzugmilch. Higher percentages of some pathogens were detected in raw creamery milk, farm bulk tank milk, and pasteurized milk [84].

The most recent study of US outbreak investigations associated with dairy [29] reported the following numbers of outbreaks, illnesses, hospitalizations, and deaths, and the rate of deaths/1000 illnesses for both raw and pasteurized milk. Although raw milk was associated with 2 deaths in the 12 year time period assessed, pasteurized milk was also associated with deaths (4) in the US, and 4 other deaths associated with pasteurized milk were reported from an additional outbreak in 2016 in Canada [82].

Whitehead and Lake [29] determined that outbreak rates for raw milks, adjusted for state populations and numbers of licenses over these 12 years, were not increasing with increasing access. Work on extending the analysis of Whitehead and Lake [29] is underway, considering more recent CDC data (2005–2019) and applying additional statistical techniques for specific hypothesis testing: graphical analysis by state and year, rank-sum test, and multilevel Poisson or negative binomial regression modeling.

In addition, aspects of the study by Loss and colleagues [74] are also relevant for the attenuating risk argument. The authors reported crude odds ratios for cold or runny nose (rhinitis), respiratory tract infections (RTIs), cough, and ear infections (otitis), for 983 infants in the PASTURE birth cohort study, adjusted for breastfeeding and other potentially confounding factors (including birth mode, visits to stable or livestock area, contact with domestic animals). Significant protective effects against infections were comparable for raw breastmilk and raw bovine milk (crude and adjusted) compared to pasteurized and ultra-high temperature (UHT) milks. The authors noted that raw milks may provide similar anti-infective properties. Reported significance of adjusted odds ratios for raw bovine milks included (i) rhinitis (aOR 5 0.71 (0.54–0.94), $p = 0.015$); (ii) RTIs (0.77 (0.59–0.99), $p = 0.045$); (iii) otitis (0.14 (0.05–0.42), $p < 0.001$); and (iv) fever (0.69 (0.48–1.01), $p = 0.058$). For pasteurized shop milk, the associations were not significant except for with fever (0.69 (0.48–0.98); $p = 0.038$). No clear associations of milk consumption were noted for diarrhea. Industrially processed (shop or pasteurized) milk was not protective against rhinitis, RTIs, or otitis by either crude or adjusted odds ratios. All milks except UHT milks exerted an independent protective effect on fever. This longitudinal study demonstrated significantly lower rates of infections (rhinitis, RTIs, and otitis) for childhood consumption of raw not pasteurized milks, and no increased rate of diarrhea was observed in children who consumed raw milk.

The QMRA conducted jointly by the US Health and Human Services/FDA and the US Department of Agriculture/FSIS [67] estimated that listeriosis risks for pasteurized

and raw milks were both high and of similar orders-of-magnitude. For example, the estimated annual risks associated with pasteurized and raw milks were 90.8 and 3.1 serious illnesses/deaths, respectively, corresponding to risks per serving of $1 \times 10^{-9}$ and $7 \times 10^{-9}$, respectively [67] (Summary Table 4, Figure 1). Almost a decade later, independent researchers [69] updated the 2003 FDA/FSIS [67] QMRA for listeriosis in raw milk, and their reassessment reported very low risk (approximately $2 \times 10^{-15}$ per serving), roughly 3.5 million times lower than the 2003 QMRA suggested. More recently, another independent academic study by Stasiewicz and colleagues [70] reported increasing growth rates of the enteropathogen *L. monocytogenes* with increasing thermal processing of raw bovine milk. A subsequent study [99] reviewed data and models for listeriosis since the 2003 report and concluded that the non-threshold, low dose-linear Dose–Response model may be inappropriately conservative. Thus, inappropriate assumptions about prevalence, levels, growth rates, and Dose–Response models for the pathogen in raw and pasteurized milks contributed to positive bias of risk estimates for raw milk.

*8.3. Risk–Benefit Conclusions*

Overall biological benefits associated with raw bovine milk are consistent, with supplemental studies on plausible mechanisms attributed to biologically active raw milk based on large human cohort studies, including longitudinal studies and a recent double-blind randomized trial.

Evidence for assessing risks of foodborne infective illness from consumption of raw milks is limited based on QMRAs that included sparse data on potential pathogen frequencies, levels, growth, and dose–response relationships for raw bovine milk and many unvalidated assumptions. Effects of diverse production and management systems such as farmer training in HACCP and Test-and-Hold programs were not precisely simulated in the available QMRAs. Nor were data at this granularity (farm-level practices, pasture grazing vs. confined feeding operations, etc.) available in systematic reviews, meta-analyses, or cohort studies. Further, the evidence from a large longitudinal cohort study (Loss et al., 2015) contradicted findings from QMRA studies and prior observational studies (all type 2 outbreak studies without denominators, inappropriate for demonstrating causality as per Jaros and colleagues [68] that was cited by Whitehead and Lake [29].

*8.4. Remaining Uncertainties*

Although plausible evidence exists for multiple mechanisms likely to contribute to health benefits associated with raw bovine milk, further research is needed to deepen understanding of the dynamics and functionality of bioactive components of the raw bovine milk ecosystem and thermally treated milks, including milk powders, so that benefits, particularly to malnourished and stunted children around the world [100], can be maximized.

Future research needs for the raw bovine milk ecosystem are noted below.

- Are the presence and level of potential pathogens in raw milk predictive of risks (illness)? Are the presence and levels of the natural milk microbiota (or smaller consortia) predictive of benefits (protection against infectious illness and NCDs)? Do we need metrics from monitoring of both potential pathogens and the core consortia of the milk microbiota to assess the balance of benefits and risks?
- Is the assumption that risk (likelihood and/or severity of infectious illness and NCDs) to children drinking raw milk is higher compared to adults supported by current evidence and analysis?
- Is 'zero tolerance' for pathogens (or their toxins) in raw milks scientifically, economically, and ethically defensible, given current evidence and analysis?
- Who benefits from access to raw, pasteurized and dry milks?
- What level of risk reduction can be achieved by HACCP programs, cold chain, and other farm management practices that maximize herd health and minimize: (i) fre-

quency and duration of mastitis; and (ii) frequency and level of contamination by potential pathogens in raw milk from farm to table?

## 9. Opening Dialogue and Future Directions

The evidence maps for the bovine milk ecosystem generated herein (Figure 1) and the breastmilk ecosystem in a companion article [64] illustrate the major shifts in methods, concepts, and knowledge base [101] that microbiologist and physician Martin Blaser [102] described as the 'microbiome revolution'. Our evidence maps highlight plausible mechanistic studies on the microimmunosome and the natural microbiota of raw milks that may contribute to benefits and risks.

Both for raw and pasteurized donor breastmilk and cow's milk, no comprehensive application of formal risk–benefit methodology informed by transdisciplinary knowledge of the natural microbiota of milks was identified in our literature searches to date. Yet, opinions that risks exceed benefits for raw milks can be found in review articles [103,104] and a recent workshop paper [105]. Although the workshop paper by Verhagen and colleagues [105] presented their work as an exploratory approach to raw milk risk–benefit analysis, the case study considered only enhanced vitamin B2 intake as a potential benefit in relation to listeriosis risk. Verhagen and colleagues [105] stated that additional risks and benefits including potential protection from autoimmune diseases and the effects of the microbiota of raw milks should be considered in future expansions of their exploratory work. We note that the workshop paper did not cite any of the studies included in the Evidence Basis of the bovine milk evidence map (Figure 1). The two evidence maps for breastmilk and bovine milk, particularly the intersections of immunology and microbiology, challenge oft-stated opinions that pasteurized milks are more beneficial and less risky to consumers, opinions that appear to be firmly rooted in outdated assumptions from the pre-microbiome world.

Two recent reviews [28,84] noted links between the organic and raw milk movements. Both cited the work of Lady Eve Balfour, founder of the Organic Movement and Soil Association in the UK, and her 1948 perspective that pasteurization was a 'confession of failure' that should be abandoned as soon as sanitation issues in urban and rural dairy operations could be controlled. Lady Eve Balfour's work on whole foods included raw milk traditionally produced on holistic small farm ecosystems and identified factors relevant to future risk–benefit analysis. Optimistically, the evidence maps for breastmilk [64] and bovine milk (Figure 1) may serve to re-open deliberations of the 'state of the science' and uncertainties about benefits and risks of pasteurization in the 21st century.

### 9.1. Updating Preconceived Notions from 20th-Century Decision Science

The need for trans-disciplinary analysis that structures whole bodies of evidence for benefits and risks of raw and pasteurized milks is urgent, as restoring health may be tied to restoring diverse microbes to the industrial diet dominated by processed foods, including pasteurized milks. From our perspectives, the proposal of considering Recommended Daily Allowances (RDAs) not just for vitamins, but expanding RDAs for daily doses of microbes [35] has merit. Our work emphasizes decision science, immunology, medical microbiology, and their intersections for the bovine milk ecosystem summarized in the Evidence Map (Figure 1). From our point of view, attention to these disciplines, as well as statistical characterization of causality, is crucial for opening public discourse about benefits and risks for raw and pasteurized milks beyond discussions historically focused solely on outbreaks.

Decisions to prohibit access to raw bovine milk (e.g., in some US states, Australia, Canada, Scotland, and potentially Ireland) or recommend boiling raw milk (e.g., New Zealand, UK, and others) can no longer be claimed as evidence-based public health protection, given what is known of the microbiota of milks (Figure 1) and evidence of interactions with the microimmunosome and host immune, gut, respiratory, and neurological systems. Certainly, large bodies of evidence on benefits and risks exist, and plausible mechanisms are

consistent with benefits and risks in breastmilk and bovine milk consumers (Supplemental Tables S1 and S2), though incompletely characterized.

Data are needed documenting Exposure Assessment and Dose–Response Assessment, use of plausible alternatives to the traditional conservative non-threshold dose–response models that ignore principles of microbial ecology and innate protections of healthy gut microbiota with colonization resistance [98,99,106,107]. Data on effectiveness and costs of alternative risk reduction strategies are also needed, as well as data on efficacy of thermal and non-thermal treatments for bioactive milks.

The USDA Economic Research Service routinely collects data on estimated consumption of diverse pasteurized dairy products, including fluid milks, based on retail sales data for 'disappearance' of barcoded products from retail shelves (https://www.ers.usda.gov/data-products/dairy-data/ (accessed on 15 January 2021). No data are collected by this or any other US government agency for raw milk, limiting evidence-based Exposure Assessment for raw milk. Such data could also be generated for retail raw milk in the 10 US states that currently permit retail sale of raw milk (CA, CT, ID, ME, NH, NM, PA, SC, UT, WA), marked by green fill in the map below (Figure 2).

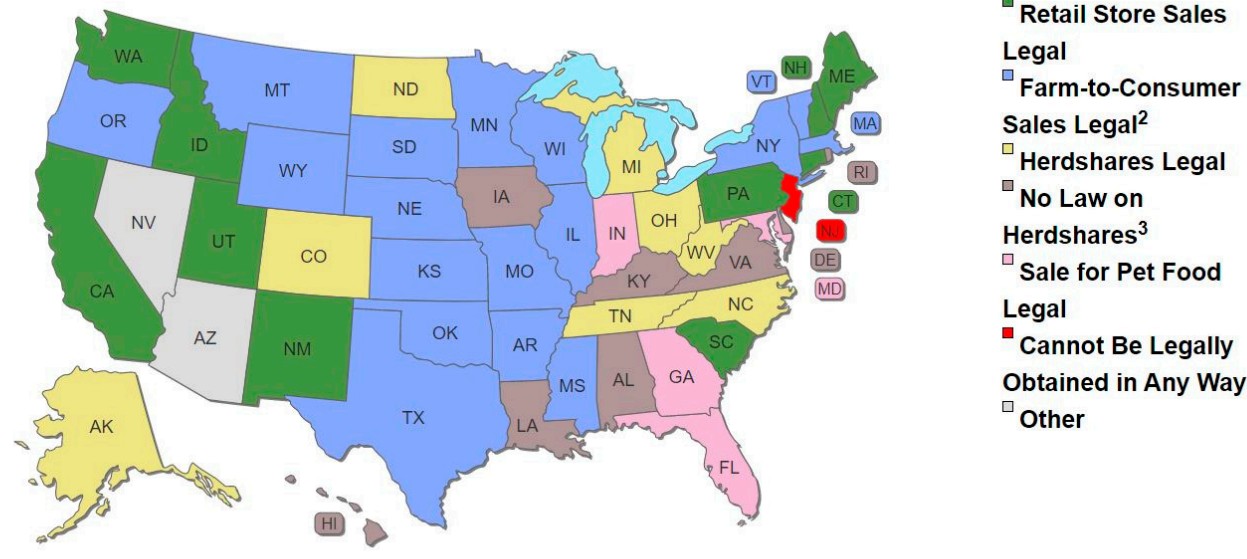

**Figure 2.** Map of Legal Status for State Raw Milk Laws as of August 2021 (used with permission from Farm-To-Consumer Legal Defense Fund; for additional information on specific state laws, see table at https://www.farmtoconsumer.org/raw-milk-nation-interactive-map/, accessed on 2 November 2021).

We note that one might expect that benefits and risks may differ by age and health status of consumers, estimated daily consumption of raw milks and other foods, as well as state or region, year of production, and legal status for access to raw milks (retail, on-farm, herdshare, pet food, prohibited). However, additional information would be needed to estimate benefits and risks across these variables. Although significant data gaps likely limit quantitation of benefits and risks for dietary exposures for raw milks and other foods, the qualitative evidence mapping approach applied herein has utility for informing future research and facilitating future evidence-based decision making for raw and pasteurized milks.

Further, data and analysis are needed to evaluate the effectiveness of current 'zero-tolerance policies' for potential pathogens in raw milk (presence of pathogens at any level constitute adulteration, unfit for human consumption) in many US states and countries around the world. Decisions about assessing and managing benefits and risks based on presence/absence data or genus-level information on potential pathogens are increasingly

recognized as problematic. Use of non-pathogenic strains related to pathogens, such as *Clostridium* and other gut commensals protecting against *C. difficile* and *Staphylococcus* spp. protecting against *S. aureus* [97] are consistent with Microbiome-First approaches. Application of next generation microbial ecology and multiple interdependent mechanisms could prove useful in designing therapeutics to re-establish a healthy microbiota and restore colonization resistance against nosocomial [36,37] and food-borne pathogens [38] in the future.

Considerable evidence is available from the discipline of predictive microbiology that defines growth limits for potential bacterial pathogens at temperatures typical of refrigeration and temperature abuse [98] that are crucial to consider along with different growth limits of the dense, diverse natural microbiota of milks. The few QMRAs available that estimate risks appear to grossly overestimate risks and underestimate uncertainties for raw milk consumers by applying a series of worst-case assumptions. The EFSA (pg. 4) [71] observed the following, questioning the need for zero-tolerance policies for listeriosis in monitoring programs for raw milk.

Although *L. monocytogenes* is not considered to be one of the main hazards associated with RDM (raw drinking milk) in the EU, the reviewed QMRAs from outside the EU do show that the risk associated with *L. monocytogenes* in raw cow's milk can be mitigated and reduced significantly if the cold chain is well controlled, the shelf-life of raw milk is limited to a few days and there is consumer compliance with these measures/controls.

Overreliance on poorly characterized epidemiologic associations from case studies and outbreak investigations on raw and pasteurized milks are problematic for many reasons, particularly exaggeration of risk and underestimation of uncertainty based on sparse, weak correlative data and a paucity or lack of statistical analysis [68]. Davis and colleagues [103] noted that comprehensive simultaneous evaluation of benefits and risks is crucial to support the positions that risks outweigh benefits (or benefits outweigh risks), and no such study has yet been published for raw milk, to our knowledge. Our intent in this work is to 'let the data (and analysis) speak' and convene stakeholders to objectively and transparently evaluate benefits and risks simultaneously using established methods for risk–benefit assessment [65,108,109].

A recent report [90] cited data for production of more than 1,352,000 gallons of fluid raw milk provided to retail markets in California from Organic Pastures LLC (Fresno, CA) from 2018 to 2020. For a typical serving size of 250 mL, this volume of retail raw milk would represent approximately 20,480,000 servings. Given that no outbreaks of illness associated with raw milk were reported in California in that period, the data are consistent with a risk of illness less than 1 in over 20 million servings for retail raw milk consumers.

Work that bridges the epidemiology and risk assessment gaps for microbes, as recently addressed for chemical hazards [110], would likely also enhance transparency in public discourse, holistic or trans-disciplinary education, and public health decision-making regarding raw and pasteurized milk issues. Our evidence map work is intended to increase transparency about the recent bodies of evidence regarding the controversial issues around milk pasteurization and more importantly, to improve transparency in public discourse about the evidence on raw milks, promote research to fill data gaps for risk assessment, and enable development of evidence-based regulations and policies that balance benefits and risks in the near future. Together, these two evidence maps are envisioned as the foundation, the scientific basis, for an international workshop to launch a series of exercises of analytic-deliberative process [111,112] around the world planned for this project in future years.

Risk managers of the 21st century may now choose to consider opportunities to update the science base for testing and regulation of foods, particularly foods such asraw milk containing a dense and diverse natural microbiota that competes with pathogens and supports GI, immune, neural, and respiratory system health. Clearly, such opportunities will require a major paradigm shift in the US and around the world from current zero-

tolerance policies for presence of potential pathogens in raw or ready-to-eat foods that rely on outdated 20th-century science.

### 9.2. Updating Preconceived Notions from 20th-Century Microbial Ecology

The microbiota of milks is influenced by many factors (Supplemental Section A), and the 20th-century notion that the microbiota are simply fecal contaminants posing high risk to human health seems invalid. Perceptions of bacteria as germs to be eradicated are gradually being replaced by deeper awareness of symbiotic (commensal and mutualistic) microbiota as our partners in health [33,113].

Multiple lines of evidence support the plausible existence of an entero-mammary pathway for transfer of microbes or their DNA from the maternal gastrointestinal (GI) tract to mammary tissue and subsequently to milk and the oral cavity and GI tract of breastfeeding infants [12,114]. In addition, these authors consider conflicting evidence for the infant oral (bucchal) and the maternal skin microbiomes as potential sources of microbes in breastmilk. However, the authors cite six studies documenting transfer of a total of 14 microbes to date from maternal GI to breastmilk and to infant GI. Consistent with Zimmerman and Curtis, additional reviews [115,116] conclude that the predominance of available scientific evidence supports the entero-mammary pathway of transferring maternal GI microbes to breastmilk and breastfeeding infants. The review by Oikonomou and colleagues [10] cites some of this evidence and one study [117] supporting the existence of a homologous entero-mammary pathway in bovines and concludes that the body of evidence suggests transfer of microbes from milk to infants via an entero-mammary route.

A small but elegant study conducted by Wu and colleagues [94] explored relationships between microbiota of the bovine rumen and GI tract (feces), milk, and the cowshed environment (airborne dust, bedding, feed, water). The cows were housed in freestall barns (not pastured) and fed mixed ration silage. Samples were analyzed by quantitative real-time PCR to the bacterial family level for major taxa (present at >1% in at least two different samples). Results comparing total population and bacterial composition between two farms were assessed by analysis of variance. Analysis of the relationships between potential sources of microbes was conducted using a published SourceTracker algorithm, hierarchical clustering and heat mapping, and canonical analysis of principle coordinates methods. The bullets in Table 2 provide perspective on this innovative research, as well as surprising results that challenge prior assumptions about the milk microbiota. For example, the gut-associated microbiome assessed from fecal samples was not a primary risk factor for predicting contamination of milk or for bovine mastitis on the two farms studied.

Wu and colleagues [94] concluded that the raw bovine milk microbiota is clearly separated from the fecal microbiota (as well as the microbiota associated with feed, rumen fluid, and water). This finding contradicts the common notion that bacteria present in milk are fecal contaminants. Together with studies supporting the entero-mammary pathway of transfer of microbes in healthy hosts, these results challenge 20th-century notions about the milk microbiota and merit further deliberation for evidence-based decision making. Clearly, systematic research studies are needed to determine how generalizable these results are to other dairy farms, breeds, farm management systems including pasture-based herds, and other factors influencing the microbiota of milks.

The body of evidence for factors influencing the microbiota of milks related to the 'environment' aspect of the traditional 'disease triangle' (e.g., air quality and pollution; diet; supplements and pharmaceuticals; behavior/lifestyle/environment including farm and non-farm environments, built and natural environments, organic and industrial dairy practices; dust and soil; water) is extensive and relevant to modeling dose–response relationships for pathogens amidst the dense and diverse microbiota of raw foods including milks (Coleman et al., 2018) [106]. Some notable recent studies relevant for future analytic-deliberative process include the following [86,118–130].

*9.3. Updating Preconceived Notions from 20th-Century Immunology*

An influential paper [131] documented similarities in proteomes of raw human and bovine milks associated with host defense functions, including the finding that human and bovine milks shared 33 proteins of identical sequence. This finding suggests that components of bovine milks may support immune function across species, e.g., in human and other animal model systems. Other more recent evidence (Figure 1) links NCDs and acute infectious diseases with feeding pasteurized milks more often than raw, and cohort studies document loss of benefits for infants, children and adults consuming pasteurized milks.

A second key paper [132] predicted the functionality of the human milk metagenome enriched in genes for nitrogen metabolism, membrane transport, and stress response, observations that are consistent with defense functions in the proteomes of human and bovine milks cited above. Deeper understanding of the functionality of the microbiota of milks could catalyze tremendous breakthroughs in therapeutic and preventative medicine, to nurture healthy microbiota or restore dysbiotic microbiota [33].

Supplemental data are now available characterizing likely mechanisms driving infectious and inflammatory disease, including colonization resistance by direct and indirect competition of the microbiota in foods and the gut. Colonization resistance enhances health of superorganisms by: (i) outcompeting pathogens in the intestinal lumen; (ii) reducing likelihood of pathogen attachment along mucosal surfaces of the gut, respiratory, and urogenital tracts; (iii) up-shifting pathogen load requirements for disease (enhancing innate resistance against low pathogen doses); (iv) strengthening mucosal barriers against pathogenesis; and (v) optimizing immune homeostasis, balancing inflammatory processes linked with allergies, asthma, and infectious disease [33].

Although animal models have provided new insights into potential mechanisms for human immune system protections, three recent studies [36,63,124] provide both. Stein and colleagues [36] identified consortia of strains that were positively and negatively associated with *C. difficile* infection in humans and mice, described in more detail in Coleman and colleagues [97]. Abbring and colleagues [81] provide consistent evidence from murine models and a double-blind randomized human provocation pilot that demonstrate significantly greater tolerance for raw than pasteurized milks. Known volumes of raw and processed shop (pasteurized and homogenized) milks were administered orally to 11 children diagnosed with milk allergies. On average, the children tolerated only 8.6 mL of commercially processed (shop) milks before triggering allergy, while the same children tolerated the maximum volume tested (50 mL) of organic raw milk ($p = 0.0078$). The implicated mechanism was increased allergenicity of specific whey proteins by pasteurization.

The second study by Dhakal and colleagues [124] provides consistent evidence of mucosal immune maturation from a humanized germfree piglet model and from young children from five Amish (rural) and five non-Amish (urban) families. Piglets inoculated with infant fecal microbiota maintained the clear functional differences in diversity and immune effects transferred from Amish and non-Amish donors. Differences in dietary habits were determined likely to account for differences in gut microbiota, but detailed dietary information was not available. An earlier study [36] noted profound differences in 30 children from Amish (traditional farming) and 30 from genetically similar Hutterite (industrialized farming) families. Markers of immune function in children from traditional farms were associated with lower prevalence of markers for asthma and allergic sensitization, controlling for host genetics. These differences were consistent with asthma and allergy markers induced in mice exposed to house dust from respective families.

Taken together, these three studies [36,81,124] incorporating animal model systems and human subjects provide relevant evidence about plausible mechanisms for beneficial effects of the gut microbiota and diet on immune system balance for traditional farm families who are more likely to consume raw milk.

*9.4. Future Directions*

Our evidence maps for breastmilk [64,97] and cow's milk described herein reflect the explosions of knowledge of the microbiota of milks and their functions in health as the 'microbiome revolution' continues to transform into 'normal science', in Kuhn's terminology [133]. Tremendous advances in the state of knowledge about milk microbiota for both humans and cows (analysis herein) are highlighted from the past decade, from microbiologic, immunologic, and decision science perspectives. Of particular importance for future deliberations about evidence for benefits and risks is the strength of evidence for making valid statistical inferences about balancing benefits and risks while acknowledging uncertainties about likelihood and magnitude for both.

Public media and even scientific papers may focus narrowly on outbreaks of acute foodborne infections as though epidemiology alone is sufficient for risk analysis. Greater public health benefits may be associated with "complete" unprocessed foods (e.g., raw milks) that support and beneficially regulate the microimmunosome [1] and protect against NCDs, notably asthma and allergenicity [63]. Importantly, much of the epidemiologic evidence for milk risks is correlative, not causal. For example, just as children's shoe size is correlated to reading ability but is not a causal predictor [134,135], the presence of foodborne pathogens in milks may be correlated to illness, but provide insufficient evidence to predict the likelihood and severity of health risks. Clearly, trained dairy farmers in the 21st century can reliably produce hygienic raw milk for direct human consumption that rarely tests positive for pathogens (see Table 1).

As discussed previously, the blanket application of outdated dogmas from 20th-century science [66] is problematic for regulating 21st-century raw milks. Considering our understanding of the microbiome and the microimmunosome combined with today's animal health practices, we now understand that there are both benefits and risks to be weighed. Wider deliberation of the evidence of benefits and risks is needed to develop shared understanding of technologic advances in methodology and practice for microbial risk–benefit analysis. Until such efforts are undertaken, outdated notions about the microbiota of milks will deter development of evidence-based policies for raw and pasteurized milks. The evidence maps generated in our studies are anticipated to form the basis of a series of international workshops that will address the changing paradigm of milk ecosystems and initiate the first cycle of analysis and deliberation with decision makers and stakeholders in the 21st century.

**Supplementary Materials:** The following are available online at https://www.mdpi.com/article/10.3390/applmicrobiol2010003/s1, Table S1: Supplemental Studies on Plausible Mechanisms and Relevance for Pro-Argument on Bovine Milk Benefits (Figure 1); Table S2: Supplemental Studies on Plausible Mechanisms and Relevance for Contra-Argument on Bovine Milk Risks (Figure 1). Supplementary file S1: Evolving Context of Microbiota of Mammalian Milks.

**Author Contributions:** Conceptualization, M.E.C., R.R.D. and D.W.N.; methodology, M.E.C.; formal analysis, M.E.C.; investigation, M.E.C. and R.R.D.; resources, M.E.C.; writing—original draft preparation, M.E.C. and R.R.D.; writing—review and editing, M.E.C., R.R.D. and D.W.N.; visualization, M.E.C. and M.M.S.; project administration, M.E.C.; funding acquisition, M.E.C. All authors have read and agreed to the published version of the manuscript.

**Funding:** Largely unfunded project with partial support from crowdfunding.

**Data Availability Statement:** Not applicable.

**Acknowledgments:** The authors appreciate helpful comments from journal reviewers. We are grateful to Anna Ojo-Okunola (University of Cape Town) who shared a figure adapted for use in Supplemental Section A and to those individuals and organizations who contributed to a crowdfunding campaign that partially funded this work. Photographs in the graphical abstract were taken by MEC. We acknowledge assistance with graphical images by Ryan Bentz and helpful comments from Joanne Whitehead, Richard Williams, and Amy Vasquez and Janice Dietert on a previous version of this

work. We are also grateful to many members of the Society for Risk Analysis (SRA) who contributed to discussions of the evidence for this project in recent years.

**Conflicts of Interest:** Author have no potential conflict of interest to declare except M.E.C. who has provided expert testimony in several court cases regarding the microbial ecology and assessment of benefits and risks for microbial pathogens.

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
