# Peer review of "Nourishing the Human Holobiont to Reduce the Risk of Non-Communicable Diseases: A Cow’s Milk Evidence Map Example"

_2673-8007, doi:10.3390/applmicrobiol2010003_

Round 1
Reviewer 1 Report
Revue Applied Microbiology
Nourishing the Human Holobiont to Reduce the Risk of NCDs : A Cow’s Milk Evidence Map Exemple
By Dietert et al.
In this paper, Dietert et al provided an « evidence map », e,g, a type of benefit risk mind map for raw (natural) vs. processed (pateurized) bovine (cow’s) milk. The conclusions of the evidence map go towards overall biological benefits associated with raw milk and limited evidence of milk-borne risks of infectious diseases in children and adults for both raw and pateurized milks.
This is an extensive narrative on the subject (more than 130 references plus numerous complete supplemental materials), considering many aspects of the question. The « evidence map » approach is interesting as it allows a multidisciplinary approach of the question. Indeed, some context elements are missing to be able to truly appreciate the impact of such a study. The study deals with bovine milk, which is not consumed by infants (who drink breastmilk or infant formula) nor by toodlers, the recommendation being to provide low protein third age formula milk until 3 years of age if breastfeeding is not performed any more. This is only a recommendation and it would be interesting to know which proportion of toodlers do drink bovine milk (either raw or pasteurized). Furthermore, it seems that raw milk consumption is rather limited in the world but with large country differences and this will be interested have data on its effective consumption. Later in life (in adulthood), cold milk (either raw or pasteurized) is poorly consumed, yet with, once again, large country differences and such pieces of information would be interesting, in order to be able to evaluate the relevance of the present study. On the contrary, toddlers and adults eat a lot of fermented dairy products, which do represent a large amount of microbiota-laden foods, and the reason why the authors focused only on milk and not fermented dairy foods has to precised in the introduction. In this contexte, the term « processed » is ambiguous as one can think that it involves fermented dairy products : it should be removed in the contexte of this article and only « pasteurized » should be used instead. The potential for developing the consumption of raw milk has to be precised as raw milk is really a niche consumption in many countries and mainly concerns people living in or near farms.
Some parts of the article are quite long and could be reduced.
Yet, it is an extensive and complete review of the litterature dealing with the milk microbiota and its effects on various fonctions, which could be of interest for many researchers in this field.
Minor :
Title : Explicite the abbreviation NCDs as Non Communicable Diseases
Author Response
Responses to Reviewer 1:
We appreciate the review of our manuscript and the comments that were offered to improve this Paper. We are particularly pleased that the reviewer believes that the extensive materials we presented on milk microbiota are likely to be of interest to researchers and readers of this journal.
We agree with Reviewer 1 that we did not focus on diet, for infants, toddlers, or adults. We also agree that these populations may consume fermented dairy products as well as fresh unprocessed fluid (raw) milk. To address this, we altered the Abstract to make it very clear that our focus is on fluid raw cow’s milk rather than fermented dairy products (also mentioned again in the following paragraph). We also added clarifying text on this point to the body of the paper (lines 70-76 of the revision). We believe that these changes will help to provide the needed clarity and context for what we cover vs. what constitutes the totality of microbiota laden foods.
Please note that we did attempt to find relevant comprehensive, and readily presentable information comparing current milk consumption of various global populations and across age groups. We did not find ready comparisons within our tight timetable for returning this revision, and we expect that data for consumption frequency and volume will be a data gap for any countries that choose to undertake assessment of benefits and risks for raw and pasteurized milks in the future. We added clarifying text on this point to the body of the paper (lines 836-844 of the revision).
We agree with the reviewer that the submitted work had the potential for ambiguity regarding fermented dairy products. To address this, we modified, shortened, and clarified the Abstract to make it clear that the scope of our study was raw and pasteurized cow’s milk. We added text described above excluding fermented dairy products from the scope of our current work. We clarified the meaning of unprocessed in the text to convey fresh fluid raw milk that is not homogenized, pasteurized, or fermented. These changes should remove the ambiguity.
We agree with the reviewer that ‘Non Communicable Diseases’ should be fully written out in the title rather than using the NCD abbreviation. We made the exact change to the title of the article suggested by the reviewer.
The reviewer suggested that the article is quite long and that some parts could be reduced. This corresponds with reviewer #3’s mention of possible redundancy between the main and supplemental parts of the manuscript. We identified the redundancies and made these deletions. Therefore, we used this elimination of redundancy between the main text and supplemental materials as a strategy to simultaneously address the suggestions of both reviewer 1 and reviewer 3. To this end, the revisions resulted in a shortening of the Appendix A section by more than 300 words. The shortening of the Supplemental Tables was even more significant with a reduction of more than 7,000 words.
Thank you for the specific suggestions that have significantly improved this manuscript.

Reviewer 2 Report
the paper is not divided into materials and methods etc.
The results are not clearly expressed and there is no research design and there is no statistics. This research is probably better displayed as a poster than published as a paper
Author Response
Responses to Reviewer 2:
Thank you for reading our submission and offering comments to improve the work. We also appreciate the reviewer noting a concern about the organization of our work vs that of a standard experimental research paper.
We absolutely agree with Reviewer 2 that our work is not the traditional microbiological bench research experimental paper. This is because the paper is transdisciplinary and brings together different disciplinary researchers that all too rarely co-publish. Because of this, the “research” component of this paper is the application of specialized tools from risk analysis to microbiological and other evidence. Because the research is not bench-type experimental work, it did not lend itself to the typical organization of experimental research format sections (e.g., materials and methods, statistical analysis.) With this in mind, we applied the risk analysis research tools (e.g., the evidence map decision map application and analysis) to the extensive microbiological and other data base producing what should be a first-time, trans-disciplinary research analysis for this topic. Our basis for this risk analysis research consisted of more than 130 cited studies.
This specific research tool from the risk analysis discipline is supported by the references we cited (Wiedemann et al., 2011; Coleman et al., 2021). Section 7, Approach for Creating the Evidence Map for Cow’s Milk, describes the approach and context for its application. In effect it serves as a form of “Material and Methods” for our novel study. We appreciate the opportunity to: 1) clarify why this is a research and analysis paper that is important for microbiota-involved decision making; and 2) explain why our research is presented in a format that deviates from that for typical bench-type experimental research but aligns with a format that facilitates the description of and presentation of transdisciplinary risk analysis research.
Thank you for facilitating improvements in our submission and for the opportunity to provide these explanations.

Reviewer 3 Report
A manuscript entitled „Nourishing the Human Holobiont to Reduce the Risk of NCDs: A Cow’s Milk Evidence Map Example” presents comprehensive evidence map and narrative for raw/natural vs. processed/pasteurized bovine (cow’s) milk.
The topic is very interesting and the publication contains a lot of information [over 130 references are discussed] but there is the impression of a chaotic manuscript writing. The transparency of the manuscript is impaired by Supplementary material - at least some of which could be found in the main manuscript. In my opinion manuscript should be corrected to make it more clear and scientific.
My other commets/suggestions:
- Abstract should be edited to make it more „clear”.
- Line 355 „The cow’s milk evidence map in Figure – includes” (which figure?)
- Supplementary Table S1a – it should not be cited whole abstracts
Author Response
Responses to Reviewer 3:
Thank you for reading our submission and offering comments to improve the work. We are pleased that you find the work comprehensive and the topic interesting.
Regarding the impression that the writing is chaotic, we can see this issue and have attempted to provide an improved merger of the inherently transdisciplinary materials. However, we understand that the inherent blending of diverse disciplinary elements each having its own set of tools, technical terms and even jargon makes it challenging to maintain the flow typical of straightforward single discipline papers.
To address this flow issue, we focused on the possible redundancy mentioned between the main section, Appendix, and the Supplemental Materials. We believe that this may have contributed to chaos/flow impressions. We have specifically modified and deleted numerous sentences in the Appendix and Supplemental Materials to minimize any “chaotic” noise and increase the readability of the overall manuscript. Please note that the reduction of overlapping material was specifically mentioned by the reviewer. Therefore, this should be a step in the correct direction though which we address suggestions from Reviewers 1 and 3. An added benefit was that in the process of reducing the overlap within the manuscript and deleting some material, we shortened the overall word count for our revision.
We also revised the title for Section 9 and moved our initial Section 10 as a subset of the revised Section 9 title. We believe that tightening and clarifying the organization of the sections further reduces the impression of chaos. These revisions in structure also address a recommendation made by reviewer #4. This should help with the specific flow of the manuscript.
Reviewer 3 made three specific numbered suggestions. Each of these has been fully addressed by the following manuscript revisions.
- We reviewed and reworked the Abstract in accordance with the comments of all reviewers. Our modifications better define the exact scope of this paper. They also delineate the novel trans-disciplinary analysis developed within the paper.
- On line 355 of the initial submission (line 362 of the revision), we corrected the omission of a figure designation by inserting Figure 1 to specify which figure was being referenced.
- Regarding the Supplemental Table, we accepted the reviewer’s recommendation and deleted the full abstracts and replaced them with links to the papers and Abstracts, with adjustments to the column heading. These changes should remove the concerns of the reviewer. The changes have the added benefits that the section was shortened by more than 7,000 words and the full text Abstracts can still be readily accessed by the readers via the links.
Thank you for greatly enhancing this manuscript with these suggestions. Through the indicated changes, we have attempted to address each and every suggestion made by Reviewer 3 and all other reviewers.

Reviewer 4 Report
pdf file

Author Response
Responses to Reviewer 4:
Thank you for taking time to carefully read and reflect on our submission. We agree that it represents detailed and extensive literature reviews in diverse disciplines. Certainly, our intent was to characterize the current state of knowledge, in all its nuance and complexity, as well as gaps in knowledge, that are essential to inform future research and facilitate future evidence-based decision making.
Regarding the suggestion by Reviewer 4 to revise the title for the Discussion section, we agree that a more descriptive title is warranted.
Note that we also considered other reviewers’ comment concerning the breadth of the material, the unfamiliar organization of the submission, and opportunities to improve clarity.
To address these in combination, we have made the following changes. 1) The title for Section 9 has been modified and expanded. 2) We moved the previous Section 10 to the subsection level (new Section 9.4). From our perspective, these changes simplify the organizational structure, highlight a different organizational pattern for this benefit-risk analysis from the typical organization of a simple experimental study, and emphasize both uncertainties about the state of the science and the need to apply the available science for future evidence-based decision making.
These changes simultaneously address overlapping concerns of multiple reviewers and should enhance the manuscript.
Thank you for the comments and for facilitating improvements to our paper.
